# Microglial Transforming Growth Factor-β Signaling in Alzheimer’s Disease

**DOI:** 10.3390/ijms25063090

**Published:** 2024-03-07

**Authors:** Natascha Vidovic, Björn Spittau

**Affiliations:** Anatomy and Cell Biology, Medical School OWL, Bielefeld University, 33615 Bielefeld, Germany; natascha.vidovic@uni-bielefeld.de

**Keywords:** neurodegenerative diseases, Alzheimer’s disease, amyloid beta, microglia, TGFβ, hippocampus, APOE, TREM2, transcriptomics

## Abstract

Novel technologies such as single-cell RNA and single-nucleus RNA sequencing have shed new light on the complexity of different microglia populations in physiological and pathological states. The transcriptomic profiling of these populations has led to the subclassification of specific disease-associated microglia and microglia clusters in neurodegenerative diseases. A common profile includes the downregulation of homeostasis and the upregulation of inflammatory markers. Furthermore, there is concordance in few clusters between murine and human samples. Apolipoprotein E, which has long been considered a high-risk factor for late-onset Alzheimer’s disease, is strongly regulated in both these murine and human clusters. Transforming growth factor-β plays an essential role during the development and maturation of microglia. In a pathological state, it attenuates their activation and is involved in numerous cell regulatory processes. Transforming growth factor-β also has an influence on the deposition of amyloid-beta, as it is involved in the regulation of key proteins and molecules. Taken together, this review highlights the complex interaction of apolipoprotein E, the triggering receptor on myeloid cells 2, and transforming growth factor-β as part of a regulatory axis in microglia at the onset and over the course of Alzheimer’s disease.

## 1. Introduction

Neurodegenerative diseases (NDs) sharing characteristics of aggregated and misfolded proteins, including but not restricted to Alzheimer’s disease (AD), belong to the group of protein-misfolding disorders [1]. In the case of AD, the amyloid-beta (Aβ) peptide as well as hyperphosphorylated Tau proteins tend to form aggregates and plaques in and around neuronal cells. Tau is a microtubule-associated protein that, if hyperphosphorylated, detaches from the microtubules, thereby leading to their disintegration and the accumulation of neurofibrillary tangles (NFTs) inside neuronal cells [2]. Aβ is a peptide 37 to 43 amino acid long derived from the β-amyloid precursor protein (APP). Under physiological conditions, APP is mainly cleaved by α-secretase first and then by a γ-secretase, resulting in soluble APP fragments that are secreted into the extracellular space and degraded after completing their function (the non-amyloidogenic pathway). Mutations of the secretases can lead to the enhanced production of Aβ by favoring the proteolytic processing of APP by β- or γ-secretase. Enhanced cleavage of APP by β- and γ-secretases leads to the formation of an amyloidogenic Aβ fragment that tends to form aggregates and can be found in the extracellular deposits (the amyloidogenic pathway) [3,4]. Aβ is found in the cerebrospinal fluid (CSF), plasma, and in almost all peripheral and brain cells. The physiological function of APP and Aβ is not clear; however, evidence points towards a possible protective role of Aβ in normal neuronal function and synaptic communication [5]. Its oligomeric form can interact with synapses, thereby altering their function. Except for having a genetic predisposition, aging should be considered the most important risk factor for developing AD, although the fundamental causes behind this disease are still under great debate [6,7]. Ever since the amyloid cascade hypothesis was introduced, most research has focused on the deposition of amyloid-beta oligomers [4]. The neuropathological features of AD and the pathways that lead to the generation of various Aβ species are summarized in Figure 1. Aβ, in its monomeric form, is thought to be non-toxic; however, aggregated forms of Aβ are supposed to be neurotoxic, with several mechanisms of toxicity. The binding of Aβ oligomers to cell surface receptors can lead to the disruption of various cell signaling pathways. Among other things, key cell functions such as calcium homeostasis are impaired, reactive oxygen species (ROS) are formed, and mitochondrial and synaptic dysfunction are triggered. In addition, the integrity of the cell membrane itself can be altered in terms of increased permeability and the formation of Aβ-channels (reviewed in [8]). 

In addition to the various forms of amyloid toxicity, more attention is being paid to inflammatory processes as a cause of the development or at least the progression of this type of dementia [9,10]. Inflammation of the central nervous system (CNS), referred to as neuroinflammation, is mediated by the activation of the brain’s innate immune system in response to inflammatory challenges also including accumulated proteins [11]. 

One aspect of the neuroinflammatory process, seen in normal aging as well, involves the increased permeability of the blood–brain barrier (BBB), allowing monocytes from outside to infiltrate the brain [12]. These peripheral immune cells become activated and release cytokines and signaling molecules that will either be protective or detrimental to the neural milieu. For a long time, it was believed that the brain’s immune system is unique and separate from the periphery. Actually, the CNS owns a special form of defense mechanisms against pathogens and invaders that is different from the innate or adaptive immune system. Important players in this defense system are glial cells. Glial cells are non-neuronal cells, subdivided into microglia, astrocytes, and oligodendrocytes, with different modes of action and function. Astrocytes and microglia actively take part in the defense process, whereas oligodendrocytes have a more supportive function by building up the myelin sheath around axons. Astrocytes and microglia secrete cytokines and growth factors to support the growth of neurons and maintain their function. Cytokines released by activated glial cells, in turn, can activate other glial cells in a self-perpetuating cycle. The cytokine profiles of microglia and astrocytes are highly similar. Both cell types are capable of secreting cytokines, chemokines, and other soluble molecules that regulate inflammatory processes in the CNS [13,14]. Typical pro-inflammatory cytokines produced in response to Aβ oligomers are interleukins (IL)-1α and β, IL-6, and tumor necrosis factor (TNF)-α [15,16]. On the other hand, glial cells are also known to secrete anti-inflammatory cytokines. These include IL-4, IL-10, IL-12, and the transforming growth factor (TGF)β-family [17]. A more detailed overview of the occurrence and expression levels of cytokines during the course of mild cognitive impairment (MCI) and AD is given in [18]. Cytokines and chemokines are actually reliable parameters to measure inflammation in the CNS because typical markers of inflammation such as swelling and rubescence are missing in this context [19]. Nevertheless, cytokine measurements also bear difficulties. The difficulties regarding general cytokine measurements are that there is still a lack of assays detecting very low levels of soluble cytokines in the CNS, and it is not easy to differentiate between cytokines expressed by different cell types [20]. There are opposing opinions about the contribution of the different cell types at the beginning and during the progression of the disease. One major obstacle in defining the role of distinct cell types during disease progression is that there is no gradual progression and no clear cut between the different stages of the inflammatory process that is ongoing in AD brains. To distinguish between an acute phase and a chronic phase of AD is still difficult, as it is difficult to differentiate which cells are specifically active in which part of the reaction and in which brain region, as the deposition and ongoing neurodegenerating processes are not synchronous between brain areas. Another factor is that there are almost no human studies regarding the very early stages of the disease, because further examinations are mostly made after clinical symptoms have become manifested. Most patients are diagnosed with AD years after the molecular processes have already begun. Especially regarding the non-inherited (sporadic) form of AD, it is still impossible to exactly mark the beginning of this disease as its causing factors are unknown. 

Next to cytokines and chemokines, molecules like nitric oxide (NO) and free oxygen radicals serve as markers of inflammatory processes in the CNS. NO and ROS are produced during the phagocytosis of cellular debris by microglia. Oxygen radicals contribute to lipid peroxidation, thereby damaging cells and DNA; they are capable of altering proteins in a toxic way, take part in neurotoxicity, and can induce apoptosis [21]. Furthermore, the neuronal- and astrocytic-derived inducible nitric oxide synthase (iNOS) levels and iNOS-positive cells are significantly higher in patients with AD than in controls [22,23]. Together, these results could explain the sustained oxidative damage over the course of this pathology, since Aβ has also been found to be a potent inducer of NO itself [24]. Astrocytes and microglia are in a close relation at the spatial and molecular level, since the co-culturing of microglia with astrocytes has shown that factors secreted by astrocytes can modulate microglial activation and result in a decreased production of ROS and iNOS [25,26]. 

Activated glial cells have been found around senile plaques in the brains of patients with AD, expressing various kinds of inflammatory markers (reviewed in [27,28]). The complete and complex cellular mechanisms leading to the neuroinflammation and neuronal death phenomena observed in AD are still unclear. Transgenic mice studies have revealed a crucial role for transforming growth factor-beta 1 (TGFβ1) in microglia maturation and activation (reviewed in [29]). Briefly summarized, TGFβ1^−/−^ mice show excessive microgliosis and neuronal cell death as well as systemic inflammation and premature death [30,31]. These data indicate a crucial role for TGFβ1 in microglia development and overall neuronal survival. Late-onset AD (LOAD) is mainly based on genetic mutations in apolipoprotein E (APOE), which, among others, is also expressed by glial cells in the CNS. APOE has been found to be connected to microglial TGFβ and the triggering receptor expressed on myeloid cells 2 (TREM2). Recent studies of transcriptome, reactome, and single-cell sequencing have shed light on a more complex distribution pattern of microglia in physiological aging and disease [11,32]. Understanding the multifactorial nature of AD is crucial for developing effective prevention strategies and therapeutic interventions aimed at delaying or halting the progression of this devastating neurodegenerative disorder. The new data from these studies revealed an interaction of TGFβ, APOE, and TREM2 in microglia and opens directions for future therapeutic interventions. In the current paper, we focus on the role of microglial TGFβ signaling and interactions during the neurodegenerative processes of AD by reflecting on recent transcriptomic data from mice and human studies.

## 2. The Pathogenesis of Alzheimer’s Disease

Alzheimer’s disease is the most common form of dementia (60–70%) affecting the elderly [33,34]. To date, the major risk factor of AD is aging [35]. In these times of demographic change, the numbers of patients diagnosed with AD are rising [36]. Most patients suffer from a sporadic form of AD, with no precise triggering factors. Next to AD from unknown causes, patients with familial or inherited AD (fAD) make up about <1% of all cases. Currently, there are three different genes known to be affected by mutations related to fAD. These genes include the genetic information for APP and both presenilins (PSEN1 and PSEN2), the functional subunits of the γ-secretase which is crucial for the cleavage of APP into Aβ peptide which are 40–42 amino acids long (Aβ_40_/Aβ_42_) [37]. These mutations lead to an increased production or altered processing of Aβ, resulting in the accumulation of amyloid plaques in the brain. In addition, genetic variations in TREM1 and TREM2 have also been linked to an increased risk of AD [38]. TREM2 is mainly expressed in microglial cells and involved in the regulation of inflammatory reactions in the CNS as well as in recognizing bacterial invaders [39,40]. 

Corresponding to the cognitive decline observed, there are significant reductions in neuronal cell numbers and in the thickness of the cortex in AD and in normal aging processes, revealing a similar pattern [41,42]. Yet, there are differences in the outcome of these events. AD-related cognitive decline corresponds mainly to the atrophy of distinct brain areas, different from aging [43]. The hippocampus and, thereby, the formation of short-term memory as well as long-term consolidation are impaired in the very early stages of AD. These symptoms of memory decline reflect a reduction in the number of neurons in the hippocampus and entorhinal cortex. Meanwhile, the atrophy in age-related processes is due to the shrinkage of cell bodies and not their decline in number [44]. In detail, during aging, the myelin sheath around neuronal axons, which is prominently present in the white matter and produced by oligodendrocytes, becomes weak, causing impaired signal transmission and promoting axonal and synaptic loss [45]. Throughout one’s lifespan, the myelin sheath and the synapses undergo remodeling processes and display plasticity [46]. While familial forms of AD manifest themselves symptomatically quite early on, in the late-onset type of AD, these processes can go unnoticed for decades. 

Unlike fAD, which has a strong genetic component, LOAD is thought to arise from a complex interplay of genetic, environmental, and lifestyle factors. While the exact etiology of LOAD remains elusive, several genetic risk factors have also been identified through genome-wide association studies (GWASs). These genetic risk factors contribute to the accumulation of Aβ and tau protein aggregates in the brain, leading to neurodegeneration, synaptic dysfunction, and cognitive decline. Moreover, these genetic risk factors include mutations in the genetic coding for the E4 allele of APOE [47]. Heterozygous carriers of this gene mutation have a three-fold risk of developing AD [48]. The human APOE exists in three isoforms, namely, apoE2, apoE3, and apoE4, depending on the three different alleles ε2, ε3, and ε4 [49]. APOE is a glycoprotein mainly expressed in the liver and brain by neurons, astrocytes, and macrophages and serves in the transport of cholesterol as well as in neuronal growth and repair [50]. The levels of APOE in the CSF are associated with cognitive decline in patients with AD [51]. Increased microglia reactivity has been paired with an increased expression of APOE in transgenic mouse models [52]. Recent animal studies have shown that APOE4 was increased in nuclear localization, thereby directly repressing the transcription of specific genes from a subclassification of microglia [53]. Additionally, GWASs have found specific polymorphisms in inflammation-associated genes that are linked to an increased risk of AD [54,55]. Among the identified genes is the apolipoprotein CLU. It has been found to interact with soluble forms of the Aβ species in vivo, its levels are increased in areas affected by neurodegeneration, and it has been detected in the CSF of patients with AD [56]. Both, APOE and Clu are able to suppress Aβ deposition and cooperate in modifying Aβ clearance at the level of the BBB [57,58].

## 3. Microglia Origin, Maturation, and Activation

During embryonic development, primitive macrophages from the yolk sac migrate into the developing CNS, where they undergo maturation processes and form specialized immune cells called microglia [59,60]. They contribute to the brain’s cellular content by about 5–12% [61]. Maturation and differentiation are highly regulated processes that rely on several signaling molecules such as PU.1, IRF8, IL34, CSF1, and TGFβ1 [52,60]. 

Comparable to macrophages in the periphery, microglia are capable of phagocytosis and display morphological plasticity. Under homeostatic conditions, microglia appear in to have a “ramified” shape and function as the brain’s controlling unit—scanning the surroundings for invaders, pathogens, or aggregated proteins [62,63]. In reaction to environmental changes or acute damage, they adopt different “(re-)activation” states. For a long time, the activation states of microglia have been divided into M1- and M2-like phenotypes, with differences in their morphology and cytokine profiles [64]. Reactive microglia shift towards an ameboid shape, which allows them to move freely through neural tissues and phagocytose cellular debris. Moreover, M1-like microglia tend towards a pro-inflammatory phenotype, while M2-like cells act in immunosuppression and tissue repair tasks [61]. This polarization is not final and can be altered during pathogenic progress [65]. Recently, a more diverse activation pattern of microglia that react to various “input signals” was introduced, and, at present, microglia are characterized not only on the basis of their morphology or cytokine profile but also based on their genetic (transcriptional) profile [66]. 

The input signals can have a broad spectrum and, among others, include pathogen-associated molecular patterns (PAMP), danger-associated molecular patterns (DAMP), cytokines, and growth and stimulating factors [67]. Even without DAMP or PAMP stimulation, microglia undergo morphological (hypertrophy, de-ramification, and thickening), biochemical, and transcriptional changes during aging processes, with differences across grey and white matter [68]. The transcriptome of microglia throughout the CNS is heterogenous and varies across anatomical regions, as does their distribution. Greater numbers of microglia are found in the grey compared to white matter, with the highest density being found in the hippocampus and substantia nigra [69]. Moreover, microglia expression profiles have been found to be different in the cortex and striatum compared to the cerebellum and hippocampus, with greater age-related phenotypic changes in the white matter of the cerebellum [68]. There are also regional differences in response to injury and inflammation [70].

Reactive microglia have been found in close proximity to amyloid plaques, indicating an interaction with amyloid oligomers and fibrils. Supporting evidence for an involvement of amyloid oligomers in the activation of microglia has been derived from receptors that are capable of binding Aβ_42_ [71]. Among others, the cluster of differentiation (CD) 14, which has gained importance as the lipopolysaccharide (LPS) receptor involved in innate immunity, is also expressed on the microglial cell surface, has been found to assist in the phagocytosis of Aβ fibrils [72,73]. Additionally, the receptor for advanced glycation end products (RAGE), CD36, and various scavenger receptors are able to stimulate the phagocytosis of Aβ species [74,75,76,77]. Binding to these receptors activates a signaling cascade resulting in the expression of cytokines of either pro- or anti-inflammatory origins or leads to the internalization of Aβ molecules via micropinocytosis or endocytosis. Additionally, an involvement of CD14 long time-activation by Aβ fibrils in the chronic neuroinflammatory process in AD has been hypothesized [72]. Moreover, pattern recognition receptors belonging to the group of Toll-like receptors (TLRs) are also expressed by microglia [78]. Upon activation, these receptors act via signaling induction through interferon (IFN)-γ and nuclear factor-κB [76]. TLR4 and CD14 are both involved in the recognition of bacterial LPS [79]. Like TLR2 and 4, TLR6 is also known for its Aβ-binding properties and the provoking of neuroinflammation [80,81]. CD36 acts as a coreceptor for TLR4 and TLR6, which is triggered by Aβ.

The microglial receptor TREM2 assists in the interaction with apoptotic cells, lipoproteins, and accumulated proteins such as Aβ. TREM2 is a member of immunoglobulin superfamily receptors and is involved in proliferation, survival, and regulating inflammatory processes by enhancing microglial phagocytosis. Consequently, in AD, TREM2 is increased in the microglia surrounding amyloid deposits [82]. 

## 4. The TGFβ Superfamily and Pathways

TGFβ1 is a potent cytokine that belongs to a family in the TGFβ superfamily. This superfamily consists of 33 individual cytokines that are grouped into four different main families. Those families are, namely, TGFβ, ACTIVIN/INHIBIN, bone morphogenetic protein (BMP), and growth and differentiation factor (GDF). The TGFβ family contains TGFβ1—5. In the brain, three subtypes of TGFβ (TGFβ1, TGFβ2, and TGFβ3) are present. TGFβ1 is mainly expressed by neurons and microglia in the CNS [83]. TGFβ expression in the CNS differs in a region- and cell-specific manner [84]. Pleiotropic cytokines are released in an inactive form.

TGFβ itself plays a pivotal role in the maturation of murine microglia during postnatal development, which gives rise to a specific set of genes [85]. TGFβ1 is endogenously produced by microglia themselves and secreted by exocytosis [86]. Recently, in inducible knockout models of the *Tgfb1* gene, it was shown that TGFβ1 is relatively more secreted, in an autocrine manner, by microglia than neurons or astrocytes in the adult brain to maintain homeostatic microglia [87]. TGFβ1 is crucial for the prevention of excessive microglia activation and acts as an immune suppressor [88]. 

The corresponding receptors for TGFβ1 are TGFβR1 and TGFβR2 [89]. The signaling of TGFβ is ensured via two distinct ways that have been found so far. The more complex signaling is performed through SMADs. SMADS are a family of highly conserved proteins that act as primary signal inducers for TGFβ receptors [90]. The cell membrane of microglia contains these two types of TGFβ receptors [91,92]. After binding to TGFβR2, the formation of heterotetrametric receptor complexes and TGFβ itself is induced. This formation leads to the phosphorylation of TGFβR1, mediated by TGFβR2. Receptor-activated (phosphorylated) SMADs (rSMADs) (pSMAD2 and pSMAD3) form a complex with co-SMAD4. This complex translocates to the nucleus, where it can interact with SMAD-binding elements and either recruit co-repressors (TGIF, Snon, Ski) or co-activators (CBP, p300) for the transcriptional regulation of target genes [93,94,95,96]. Those target genes, among others, code for proteins involved in ROS formation, DNA restoration, cell cycle and proliferation regulation, autophagy, aging processes, and the unfolded protein response [97]. Cell cycle modulation by TGFβ1 is the main mechanism to counteract Aβ-induced toxicity and promote neuroprotection [98]. Figure 2 displays the SMAD-dependent and -independent TGFβ signaling pathways.

Binding to TGFβR2 and forming a heterotetrametric complex with TGFβR1 can also lead to the regulation of target gene expression independent of SMADs. This bypassing route is maintained by different kinases (MAPK, MEK, AKT) and the mTOR, JNK, and p38 pathways, making TGFβ signaling more flexible with respect to environmental changes. The inhibition of receptors and rSMADs phosphorylation by SMAD6 and SMAD7, as well as γ-secretase and the metallopeptidase domain 17 (TACE), can negatively affect signal transduction and even lead to the degradation of TGFβ receptors [99,100,101,102,103]. 

## 5. TGFβ and Alzheimer’s Disease

Recent studies on the reactomes from human tissues have revealed an overlapping of 41 genes affected during aging and AD. Among others, TGFβ1 is affected in pathways including interleukin signaling and the transcription of RNA polymerase II as well as immune signaling. Surprisingly, there is also an overlap between AD and longevity genes. Longevity genes are related to a longer lifespan and delayed aging. Here, again, 43 genes overlap that involve the TGFβ1 pathways [32].

In patients with AD, the levels of TGFβ1 in the CSF and plasma are significantly higher than in the controls [15]. Interestingly, in a mouse model of AD, the reduction in TGFβ1 led to a decrease in spine density, memory function, and overall synaptic plasticity [104]. Furthermore, the loss of TGFβ signaling in the microglia was shown to result in motor deficits and impaired myelination by disturbances in oligodendrocyte maturation [52]. The knockout of TGFβ in mice resulted in severe postnatal systemic inflammatory reactions, leading to premature death and impaired homeostasis [31,105,106]. A hallmark of AD is the decreased neurogenesis of neural stem cells in hippocampal formation [107,108]. In hippocampal microglia, TGFβ has been found to be a key player in their pro-neurogenic effects in chronic neurodegeneration processes [109]. Moreover, the overexpression of TGFβ1 was even able to recover hippocampal synaptic plasticity and memory function in an in vitro model of Aβ-induced toxicity [104,110]. The parabiosis of young wildtype and old transgenic AD-mice (18 month) resulted in a significant increase in TGFβ1 levels after 3 days, and the amyloid load decreased after 14 days [111]. Interestingly, this level increase was accompanied by the increase in the cell adhesion molecule fragment L1-70 [112]. L1-70 is a cleavage product of the full-length L1, which can enter the cytoplasm and even reach the nuclei [113]. L1 itself is involved in synaptic plasticity in the traumatized and regenerating CNS, thereby supporting neuronal activities [114]. Noteworthy, the expression of L1 and L1-70 has been observed to be low in old AD-mice, among which amyloid deposition was high. On the other hand, wildtype hippocampi have been shown to contain high levels of L1-70 and no amyloid plaques. Not surprisingly, L1 has been found to bind to Aβ species 40 and 42 itself, making it an interesting candidate for therapeutic treatments [115]. 

Moreover, TGFβ1 could be involved in this process, since it has been previously found to upregulate the expression of L1 in pancreatic duct cells [116]. A knockdown of TGFβ1 in neuroblastoma cells decreased the levels of L1-70 and the pro-inflammatory cytokine macrophage migration inhibitory factor (MIF) [111]. L1-70, in turn, is able to regulate MIF expression in the brain, thereby promoting microglia activation and amyloid clearance [117]. MIF strongly is related to CD74 expression in the microglia, since their interaction leads to the activation of numerous pathways involved in cell survival and proliferation [118]. MIF has been found to be expressed by neurons rather than glial cells upon interaction with Aβ oligomers, serving as a defense mechanism, and could be useful as a potential biomarker for AD rather than MCI [119]. The studies mentioned above indicate that TGFβ1 is responsible for the expression of L1 and, subsequently, MIF through CD74. After secretion, L1 is cleaved by a serine proteinase into the L1-70 fragment, which is necessary for the expression of MIF. MIF, in turn, acts during microglia activation and promotes amyloid clearance. Since in AD mice levels of TGFβ1, L1-70 and MIF are low, there is reduced microglia activation and a higher amyloid burden, making TGFβ1 a key modulator in this system. Moreover, TGFβ1 is able to downregulate CD74 in microglia. In TGFβR2-deficient microglia, high levels of CD74 are detectable [120].

In the brains of patients with AD, TGFβR2 expression is lower than in controls without AD [121]. A reduction in TGFβ receptor genes in microglia has also been observed in aged (12–22 months) C57Bl/6J mice, with the lowest expression of TGFβR1 being recorded in the hippocampus and striatum of 22-months-old mice, whereas TGFβR2 was lowest in the hippocampus and cerebellum [122]. In 9-month-old APP/PS1 mice, TGFβR2 levels were low compared to aged controls paired with an enhanced environment of neuroinflammation [83]. The TGFβR2^−/−^ mice displayed more Aβ deposition in the hippocampus and a higher age-related neuronal degeneration rate [121]. Interestingly, in some AD cases, there is an ectopic localization of SMAD2 and SMAD3 with amyloid plaques and NFTs, thereby inhibiting its nuclear translocation and disrupting signal transduction [123,124]. Single-nucleotide polymorphisms (SNP) in the TGFβ1 gene are known to influence the expression level of the protein and thereby influence TGFβ signaling in general. In patients with AD, some functional SNPs have been found that are related to the occurrence of depression; in these cases, other SNPs were able to promote the progression from MCI to AD [125,126]. An overview of the different roles of TGFβ signaling in a healthy and AD brain is given in Figure 3. 

Though there is an increase in TGFβ1 plasma and CSF levels, there is a decrease in TGFβR2 expression in patients with AD. Enhanced expression of TGFβ1 could be a mechanism to compensate for reduced TGFβ signaling due to the reduction in receptors. TGFβ signaling is able to decrease Aβ deposition and promote its clearance if not disrupted by chronic inflammatory events [127]. 

## 6. Unique Transcriptional Profiles of Microglia during Neurodegeneration

Single-cell RNA sequencing (scRNAseq) is a revolutionary technology that enables the analysis of gene expression at the individual cell level. Traditional RNA sequencing methods provide average expression profiles from bulk cell populations, masking cellular heterogeneity and potentially obscuring rare cell types or states within a sample. In contrast, scRNAseq allows researchers to dissect cellular diversity, identify distinct cell types, characterize cell states, and uncover rare subpopulations within complex tissues or heterogeneous samples. Moreover, the scRNAseq of distinct neuronal immune cells under pathological conditions has led to the subclassification of specific disease-associated microglia (DAMs) [128]. DAMs display a unique transcriptional signature of genes that, according to GWASs, are linked to AD, amyotrophic lateral sclerosis, frontotemporal dementia, and aging [129,130,131,132]. DAMs, in the context of NDs like AD, show an upregulation of typical activation markers in correlation with a downregulation of homeostatic markers [133]. Homeostatic genes are expressed under physiological conditions and very early in the maturation process [134]. This profile includes the downregulation of homeostatic markers P2ry12, Tmem119, and Cx3cr1 [135]. Meanwhile, in DAMs, there is an upregulation of the inflammatory genes *Trem2*, *Apoe*, *Axl*, *Lpl*, *Itgax*, and *Clec7a* [131,132]. In animal models of amyloid pathology, tauopathy, and motor neurodegeneration, a correlation was found between reduced expression of homeostatic genes in microglia and the degree of neuronal cell loss. Although most of the DAM genes (*Itgax, Apoe, Axl, Cybb*) were upregulated in the animal models, the authors found little-to-no correlation with the degree of neuronal cell loss in this context, except for *Apoe*. Interestingly, in AD human tissue samples from the precuneus—a brain region which is involved in learning processes and long-term memory formation independent of the hippocampus—of 162 DAM genes, the authors found 8 genes to be downregulated, whereas the levels of prominent DAM genes (*Itgax*, *Apoe, Csf1*, *Axl*, and *Cst7)* were not downregulated at all [11]. These findings, on one hand, point towards an even more complex interaction of DAM and NDs. On the other hand, they strengthen the involvement of APOE as a DAM-specific gene in neurodegeneration. 

However, there are also more or less neurodegeneration-specific factors that can trigger microglia activation. These factors can be summed up as neurodegenerative-associated molecular patterns (NAMP) [128]. As a result, DAM can be categorized depending on their genetic expression profile, upon exposure to NAMP, as neurodegenerative microglia (MGnD) [29,136]. These genetically highly unique cells can be distinguished by the downregulation of *Cx3cr1*, *P2ry12*, *CD33*, and *Tmem119*, while *Apoe*, *Trem2*, *B2m*, and *Tyrobp* are upregulated [133,135]. The knockout of the fractalkine receptor (CX3CR1) in 5xFAD transgenic mice resulted in a ~six-fold-higher plaque burden in the hippocampus of 6-month-old mice compared to 4-month-old mice. In addition, impaired Aβ phagocytosis and clearance were observed as well as aberrant TGFβ signaling [137]. 

Furthermore, scRNAseq identified different clusters of microglia with distinct transcriptional profiles in various NDs [138]. The analysis of human microglia from the dorsolateral prefrontal cortex and temporal neocortex from autopsy samples of patients with MCI and AD and surgical interventions for epilepsy revealed 14 cell clusters. In total, 16,242 cells were analyzed, from which 99.1% were identified as microglia based on microglial-specific marker genes. Of these 14 clusters, 9 were specific for microglia. DAM genes were detected in most clusters from all the samples. According to the authors, one cluster showed a high concordance with the DAM cluster in mice. Interestingly, the pro-inflammatory marker CD74 was extensively upregulated, but the frequency was reduced in AD microglia [139]. CD74, a major histocompatibility complex class II (MHCII) invariant chain which acts in cell trafficking, was also found to be upregulated in several animal models of NDs and samples from patients with AD [132,140,141]. TGFβ1 was found to be involved in the regulation of *Cd74* expression under inflammatory conditions induced by LPS [120]. The specific inhibition of *Tgfβr1* in vitro and *Tgfβr2* in vivo resulted in the increased expression of *Cd74* in murine microglia [106,120]. To conclude, NAMP, on the one hand, act as activating factors for MGnD; TGFβ, on the other hand, is important for damping the neuroinflammatory phenotype of DAM [29]. 

## 7. APOE–TREM 2 (–TGFβ) Axis in Microglia

Overall, there is minor overlap between human and murine AD microglial gene signatures, except for APOE [140,142,143,144]. In APP-overexpressing mice, two main microglia gene clusters were identified. Cluster one lacked homeostatic microglial genes, and cluster 2 displayed the upregulation of inflammatory DAM genes. The authors of the study concluded that APOE and TGFβ are the major upstream regulators of MGnD. A specific SNP in *Tgfβ1* was found to affect the expression level of APOE4, and this SNP is over-presented in patients with AD [145]. APOE suppresses the homeostatic signature of microglia that is regulated by TGFβ [131]. Not surprisingly, the expression of APOE by microglia themselves increases with the proximity of the plaques in APP-overexpressing mice [141,146]. Although APOE induction in 5xFAD mice was shown to be TREM2-independent, genetic targeting of *Trem2* in APP-overexpressing mice suppressed APOE signaling and restored homeostatic microglia [131,132]. There is a strong correlation between TREM2 and APOE signaling. A lack of TREM2 was speculated to lock microglia in a homeostatic state, thereby blocking the defense mechanisms during AD progression [147]. *APOE* knockout studies revealed that the risk allele ε4 impairs the microglial response to neurodegenerative processes by directly repressing the transcription of MGnD genes [53]. Loss-of-function TREM2 variants in AD led to reduced APOE colocalization with amyloid deposits [148]. Overall, TREM2–APOE interaction has the potential to induce microglia activation towards MGnD (Figure 4) [131]. Understanding the intricate interplay of factors within this axis is crucial for developing effective therapeutic interventions. To delve deeper into this complex concept, it is essential to explore the nuances and interactions between the various components involved. Further investigations into the potential impact of this axis on therapeutic outcomes are necessary to elucidate its full significance. Additionally, identifying and evaluating therapeutic approaches that can modulate one or more factors within this axis will provide valuable insights into potential treatment strategies.

It is important to note that the APOE isoforms differ in their impact on the amyloid burden. Lozupone and Panza’s review of the different isoforms of APOE sheds light on the intricate role of APOE in Aβ metabolism [149]. APOE3, the most prevalent isoform, is considered the neutral or “wild-type” variant. It is associated with an average lipid metabolism and is present in approximately 60–70% of the population. On the other hand, APOE4, the isoform linked to an increased risk for AD, exhibits less efficiency in lipid metabolism compared to APOE2 and APOE3. The impact of APOE4 on lipid metabolism may contribute to the buildup of cholesterol and other lipids in the brain, ultimately leading to the formation of amyloid plaques and neurodegeneration. This differential effect of APOE isoforms on health and disease susceptibility underscores the significance of genetic variations in lipid metabolism pathways. Unraveling the role of APOE isoforms in disease risk holds promise for identifying potential therapeutic targets for conditions such as AD. The comprehensive understanding of APOE isoforms and their nuanced effects on lipid metabolism and disease susceptibility is a crucial area of study in the pursuit of therapeutic interventions for neurodegenerative diseases.

Whereas the induced expression of APOE3 in APP-overexpressing mice led to the reduction in amyloid aggregation in the cortex and the hippocampus as well as soluble Aβ_40_ and Aβ_42_, APOE4 did not have an effect on the amyloid burden in this context at all [150]. Furthermore, microglial cells are able to produce short-length Aβ species themselves during the process of clearing unfolded Aβ molecules, which can give rise to even further Aβ_42_ aggregation [151]. Microglia-derived apoptosis-associated speck-like protein containing a CARD (ASC specks) are viewed as important triggers of Aβ aggregation and seeding. Injections of ASC specks into APP/PS1 mice resulted in an increased Aβ load compared to the control mice; likewise, injections of APP/PS brain homogenates into APP/PS1 mice led to an increase in Aβ-positive deposits, whereas the injection of the homogenates into APP/PS1 and ASC^−/−^ mice did not result in any deposition. These results suggest that there is a strong correlation of ASC specks and Aβ deposition in vivo [152]. Microglia facing Aβ_42_ results in the release of ASC specks; this might be a mechanism of perpetuation and enhanced Aβ deposition. The deposition of Aβ species is thought to begin decades before the first clinical symptoms of AD arise [153,154]. Interestingly, APOE4 is known to play an essential role in this seeding process [146,155]. Moreover, in males carrying the APOE4 allele—the high-risk gene for AD—the authors of the above-mentioned study detected an upregulation of integrin beta-8 (ITGB8). ITGB8 is important in the activation of the latent TGFβ1 molecule [156]. However, in another study, the deletion of Itgfb8 was able restore the MGnD response and reduce plaque burden in AD mice, making the ITGB8-TGFβ signaling pathway a potential target for a therapeutic approach [53,157]. 

Targeting the TGFβ signaling pathway holds promising therapeutic implications in AD treatment due to its involvement in various aspects of the pathology, including neuroinflammation, synaptic dysfunction, and Aβ deposition. TGFβ signaling has been implicated in regulating the immune response in the brain, with the dysregulation of this pathway contributing to neuroinflammation, a key feature of AD pathology. Additionally, TGFβ signaling plays a role in synaptic plasticity and neuronal survival, processes which are disrupted in AD. Moreover, TGFβ signaling has been shown to modulate the production and clearance of Aβ, suggesting its involvement in the amyloid cascade. Challenges in targeting the TGFβ signaling pathway in AD treatment include the complex and context-dependent nature of TGFβ signaling, with both neuroprotective and neurotoxic effects reported in different stages of AD. Furthermore, the pleiotropic effects of TGFβ signaling in various cell types within the brain and peripheral tissues pose challenges in achieving the selective modulation of this pathway without causing off-target effects. Additionally, the BBB presents a challenge for delivering therapeutics targeting TGFβ signaling to the brain, necessitating the development of strategies to overcome this barrier. Potential strategies to modulate the TGFβ signaling pathway effectively in AD treatment include the development of selective small-molecule inhibitors or activators targeting specific components of the TGFβ signaling cascade. Moreover, approaches to enhance BBB permeability, such as the use of nanotechnology-based drug delivery systems or BBB-penetrating peptides, could facilitate the delivery of therapeutics targeting TGFβ signaling to the brain. Furthermore, combination therapies targeting multiple components of the AD pathology, including Aβ deposition, neuroinflammation, and synaptic dysfunction, may be necessary to achieve optimal therapeutic efficacy while minimizing off-target effects.

In conclusion, targeting the TGFβ signaling pathway represents a promising therapeutic approach for AD treatment, given its involvement in various aspects of AD pathology. However, addressing the challenges associated with modulating this pathway effectively, including its pleiotropic effects and BBB permeability, will be essential for the successful development of TGFβ-targeted therapeutics for AD. With the rapid advancements in imaging and sequencing technologies, researchers must stay updated with respect to the latest techniques to ensure accurate and comprehensive data collection. Additionally, it is crucial to extend research to human settings to bridge the gap between laboratory findings and real-world applications as well as consider gender-specific differences with regard to personalized medicine.

## Figures and Tables

**Figure 1 ijms-25-03090-f001:**
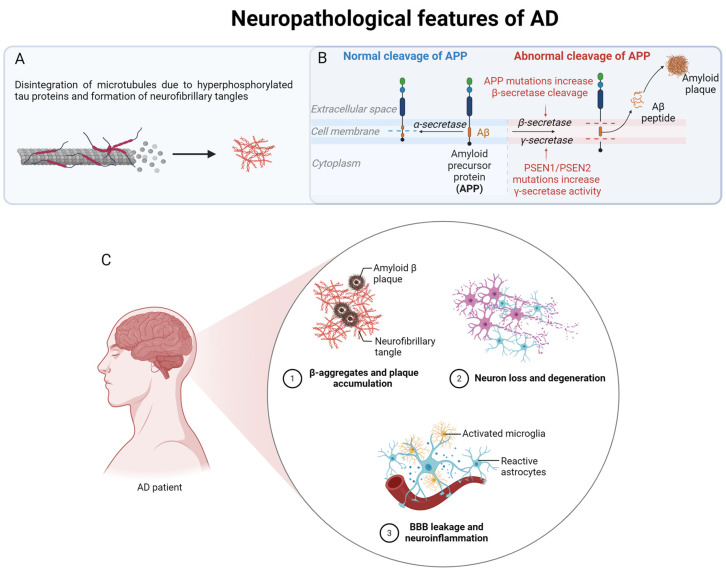
Neuropathological features of Alzheimer’s disease. Alzheimer’s disease is characterized by the accumulation of Aβ plaques outside and tau fibrils inside the neurons. (**A**) Phosphorylation of the microtubule-associated tau proteins leads to their disintegration and cytoskeletal disruption as well as to the disruption of cellular traffic and transport. (**B**) Amyloidogenic and non-amyloidogenic pathways for the cleavage of APP. APP is cleaved by α-secretase to form soluble APP fragments that are not amyloidogenic. Abnormal cleavage of APP by β- and γ-secretase leads to the production of Aβ species that tend to form aggregates. Abnormal cleavage is favored by mutations in APP itself and the subunits of the secretases. (**C**) The three main events during the pathogenesis of Alzheimer’s disease in the CNS: (1) Accumulation of plaques containing Aβ-oligomers and neurofibrillary tangles. (2) Loss of neurons and degeneration due to amyloid toxicity and NFT accumulation. (3) Activation of glial cells, release of pro-inflammatory cytokines, leakiness of the BBB, and infiltration by monocytes from the periphery correspond to main steps in the process of neuroinflammation. This figure was created with BioRender templates (https://biorender.com/, accessed on 20 February 2024).

**Figure 2 ijms-25-03090-f002:**
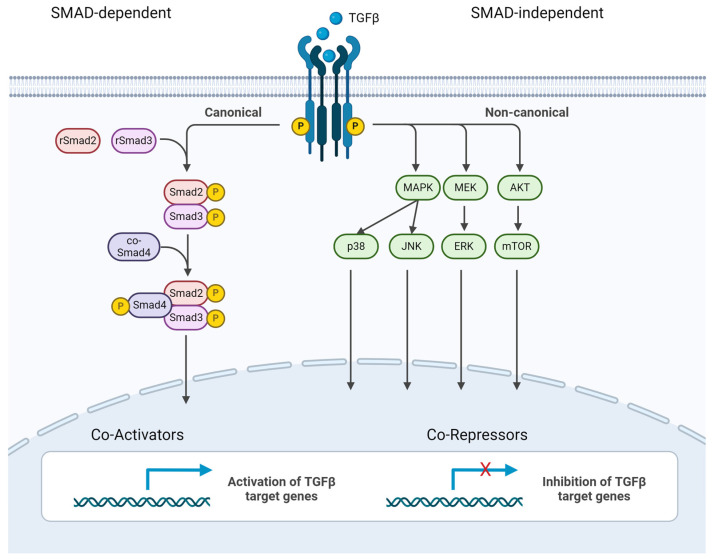
SMAD-dependent and -independent TGFβ signaling pathways. TGFβ ligands are secreted in an inactive form and become activated through proteolytic cleavage or integrin-mediated mechanisms. Activated TGFβ ligands bind to TGFβ receptors, forming a ligand–receptor complex. The binding of TGFβ ligands induces the phosphorylation of receptor-regulated SMADs (R-SMADs), typically SMAD2 and SMAD3. Phosphorylated R-SMADs form complexes with the common mediator SMAD (Co-SMAD), SMAD4. The SMAD complex translocates into the nucleus. In the nucleus, the SMAD complex interacts with DNA-binding proteins and co-activators/co-repressors to regulate the transcription of target genes, influencing various cellular processes. Binding to TGFβR2 and forming a heterotetrametric complex with TGFβR1 can also lead to the regulation of target gene expression independent of SMADs, the non-canonical pathway. This figure was created with BioRender (https://biorender.com/, accessed on 20 February 2024).

**Figure 3 ijms-25-03090-f003:**
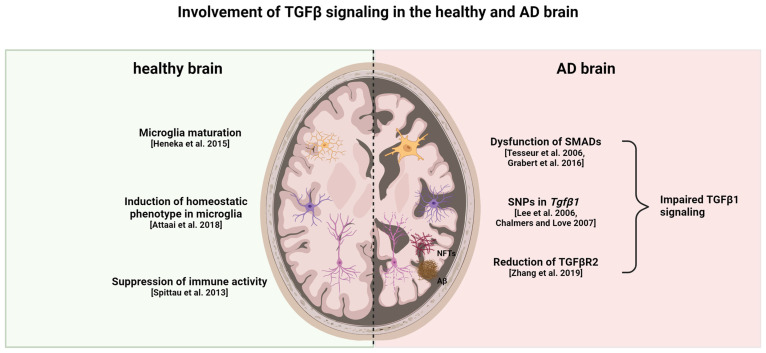
Healthy versus AD brain: differences in TGFβ signaling. TGFβ1 is important for microglia maturation and maintaining the homeostatic phenotype in healthy individuals. Moreover, it suppresses the activation of both astrocytes and microglia. Reduction in TGFβR2, dysfunction of SMAD proteins, and a specific genetic profile of TGFβ1 contribute to impaired TGFβ1 signaling in AD. This impaired signaling leads to enhanced glial cell activation in both microglia and astrocytes, thereby contributing to neuroinflammatory processes and neurodegeneration. In turn, the neurodegeneration and accumulation of Aβ and tau fibrils leads to increased gliosis and impairment of TGFβ signaling [27,85,86,119,121,122,123,124]. This figure was created with BioRender (https://biorender.com/, accessed on 20 February 2024).

**Figure 4 ijms-25-03090-f004:**
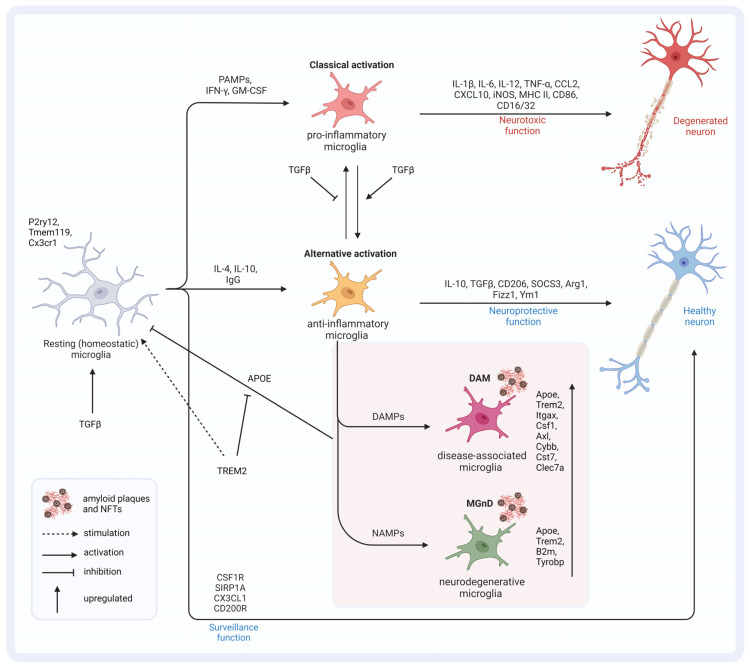
Microglia (re)activation states and the APOE–TREM2–TGFβ axis. TGFβ signaling is important for the induction of homeostatic microglia, which monitor and maintain a healthy environment for neurons. PAMPs lead to classically activated microglia of pro-inflammatory origins that exert neurotoxic function by releasing pro-inflammatory cytokines and factors. TGFβ signaling stimulates the transition to alternative activation states of anti-inflammatory origins, which exert a neuroprotective function, and inhibits the transition to pro-inflammatory cells. Depending on the signaling molecules, disease-associated microglia and microglia with a neurodegenerative phenotype can develop. Both conditions are characterized by an altered genetic profile compared to homeostatic microglia. The increased expression of APOE results in the inhibition of TGFβ-induced homeostatic microglia and favors the transition to DAMs and MGnDs. TREM2 inhibits APOE signaling and thereby favors the fixation of microglia in the homeostatic state. This figure was created with BioRender (https://biorender.com/, accessed on 20 February 2024).

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
