# Peer review of "Microglial Transforming Growth Factor-β Signaling in Alzheimer’s Disease"

_ijms, 2024, doi:10.3390/ijms25063090_

Round 1

Reviewer 1 Report

Comments and Suggestions for Authors

The review "Microglial TGFβ signaling in Alzheimer’s Disease" presents an in-depth analysis of how TGFβ signaling in microglia influences Alzheimer's disease (AD) progression. It explores the dual role of microglia in AD, highlighting their protective functions in early stages and their contribution to neuroinflammation and neurodegeneration in later stages. The manuscript discusses the transition of microglia from a homeostatic to a disease-associated state, emphasizing the role of TGFβ signaling in this process and its interaction with key AD-related genes like APOE and TREM2. Through examining genetic, molecular, and animal model studies, the review underscores the complexity of TGFβ signaling pathways in AD pathology and their potential as therapeutic targets.

The article further evaluates the implications of modulating TGFβ signaling for AD treatment, proposing that targeting this pathway could offer new strategies for mitigating disease progression. It calls for more research to better understand the nuances of microglial functions and their impact on AD, suggesting that such insights could lead to novel therapeutic approaches aimed at controlling neuroinflammation and promoting neural health. In sum, this review provides a concise yet comprehensive overview of the significance of microglial TGFβ signaling in AD, positioning it as a critical area for future study and therapeutic development.

To enhance the quality of your manuscript, it is recommended to consider the following improvements:

1. The manuscript is well-organized and presents a detailed account of the molecular pathology of AD, including the roles of Aβ and Tau, the significance of protein misfolding, and the contribution of inflammatory processes. However, it could benefit from a clearer introduction that outlines the scope of the discussion and a conclusion that summarizes the key points.

2. The manuscript references a number of studies to support its claims, which is commendable. However, it's crucial to ensure that all references are up-to-date and include the most recent research findings in the field. For instance, references to studies on the role of TGFβ signaling in microglial activation and its implications for AD could be expanded with the latest findings.

3. The manuscript provides a significant amount of technical detail, which is essential for a scientific audience. However, some sections could benefit from further explanation or simplification for clarity. For example, the discussion on the amyloidogenic and non-amyloidogenic pathways of APP processing could be enhanced with a figure or diagram to illustrate these processes visually.

4. The manuscript briefly mentions genetic predispositions and mutations related to familial AD but could delve deeper into this aspect, considering its importance. Discussing the implications of these genetic factors in more detail, including how they interact with environmental and lifestyle factors, would enrich the manuscript.

5. The section on neuroinflammation is insightful but could be expanded to include more about the dual role of inflammation in AD, highlighting both its protective and detrimental effects. Additionally, discussing emerging research on the resolution of inflammation and its potential therapeutic implications would be valuable.

6. While the manuscript does an excellent job of describing the pathophysiology of AD, it could further discuss potential therapeutic strategies or interventions that target the mechanisms described. This would provide a more comprehensive overview and highlight the clinical relevance of the research.

7. The manuscript would benefit from the inclusion of figures and illustrations to complement the text, particularly for complex mechanisms such as the formation and aggregation of Aβ and Tau, as well as the inflammatory pathways involved in AD.

8. The manuscript dives deep into the molecular mechanisms that govern microglia function and their impact on AD. However, it might benefit from a more structured breakdown of complex processes for readability. Including simplified summaries or diagrams could aid in understanding the detailed molecular pathways discussed.

9. The manuscript does well in citing numerous studies to support its claims. Ensuring that the most recent and relevant studies are referenced would further strengthen the arguments. It's also beneficial to compare and contrast different findings to provide a comprehensive view of the current research landscape.

10. The discussion on microglia's M1 and M2 activation states is insightful. Expanding on the more nuanced understanding of microglia activation beyond the M1/M2 dichotomy, considering the latest research suggests a spectrum of activation states, would enrich the manuscript. This could include detailing the various signaling molecules and environmental factors that influence these states in the context of AD.

11. The manuscript extensively covers the role of TGFβ in AD, providing valuable insights into its protective and regulatory functions. An area for expansion could be the discussion on therapeutic implications of targeting TGFβ signaling pathways in AD treatment, including challenges and potential strategies to modulate this pathway effectively.

12. The section discussing the impact of genetic variations on TGFβ signaling is compelling. Further elaboration on how these variations contribute to the risk of AD, including any potential for personalized medicine approaches based on genetic profiling, would be an excellent addition.

13. While the manuscript does an excellent job of detailing the pathophysiological mechanisms at play, a section dedicated to discussing the implications of these findings for developing therapeutic interventions could provide a more rounded perspective. This might include potential targets for modulating microglia activation or TGFβ signaling and the challenges associated with these therapeutic approaches.

14. The manuscript adeptly uses single-cell RNA sequencing data to illustrate the unique transcriptional profiles of microglia in AD. It's important to underscore the novelty and importance of these findings, emphasizing how scRNAseq has revolutionized our understanding of microglial heterogeneity in neurodegenerative diseases. Mentioning specific examples or case studies could further enrich this discussion.

15. The manuscript provides significant insight into DAMs, yet it could benefit from a more structured explanation of how DAMs differ from other microglial states, particularly in their roles during neurodegeneration. Additionally, elaborating on the implications of these findings for understanding the pathogenesis of AD and other neurodegenerative diseases would be beneficial.

16. The discussion on the APOE-TREM2 axis is compelling but could be enhanced by delving deeper into the mechanisms by which APOE and TREM2 interact and influence microglial function. The role of different APOE isoforms in modulating these processes, especially the differential effects of APOE3 and APOE4, warrants further elaboration to clarify their contributions to AD pathology.

17. While the manuscript touches on the potential therapeutic implications of manipulating the APOE-TREM2-TGFβ axis, expanding this section to discuss current research efforts, challenges, and future directions for therapeutic development based on these targets would provide a comprehensive view. Specific examples of therapeutic strategies or compounds being investigated could add practical relevance to the discussion.

18. The inclusion of Figure 2, as described, is a good practice. Ensuring that figures are clear, accurately represent the textual content, and aid in understanding complex concepts (like the APOE-TREM2-TGFβ axis) is crucial. Consider adding more figures or diagrams to visually depict the differences in microglial states, the interaction between APOE and TREM2, and the potential impact of therapeutic interventions.

19. The mention of gender-specific findings, such as the upregulation of ITGB8 in males carrying the APOE4 allele, is intriguing. Expanding on how gender differences might influence AD pathology and response to therapies could provide valuable insights into personalized treatment approaches.

20. While the manuscript does well in presenting a range of findings, a more critical evaluation of the current limitations, inconsistencies between studies, and the gaps in our understanding would strengthen the narrative. This could involve a discussion on the variability in human and animal model studies and the challenges of translating findings from mouse models to human disease.

21. Concluding with a strong section on future research directions, emphasizing unanswered questions and the next steps for investigating the APOE-TREM2-TGFβ axis in AD, would highlight the importance of ongoing research in this area. Suggestions for novel research methodologies, such as advanced imaging techniques or innovative genetic tools, could be particularly valuable.

Comments on the Quality of English Language

1. Ensure consistent use of terminology throughout the manuscript, particularly when referring to complex biological processes and molecular entities. Consistency aids in reader comprehension and maintains the professional tone of the document.

2. While the sentences are generally well-formed, there are instances where shorter or more direct sentences could enhance clarity. Complex sentences with multiple clauses can be challenging to follow, especially when describing intricate scientific processes.

3. The manuscript primarily uses passive voice, which is common in scientific writing to maintain an objective tone. However, occasional use of active voice can make the text more engaging and easier to read, particularly when describing actions taken by researchers (e.g., "The authors conclude..." vs. "It was concluded...").

4. Although the text seems largely free of typographical errors, meticulous proofreading is essential. Even minor errors can detract from the manuscript's credibility. A professional proofreading service or a careful review by colleagues could be beneficial.

Reviewer 2 Report

Comments and Suggestions for Authors

Reviewer comments and suggestions

The authors in this study discussed the complexity of different microglia populations in physiological and pathological states such as neurodegenerative diseases. The authors reported that there has been concordance in a few clusters between murine and human samples. They have included apolipoprotein E, and Transforming growth factor-β which may plays an important role in understanding neurological diseases such as in AD where transforming growth factor-β also influences the deposition of amyloid-beta. Finally, the authors summarise the interaction among apolipoprotein E, triggering receptor on myeloid cells 2 and transforming growth factor-β which may be involved in microglia functioning in causing Alzheimer’s disease.

Overall, the manuscript was well written. However, a few major concerns or comments needed to be explained or modified.

  1. Line 17 It is better to discuss your review that you can include in the text of the manuscript
  2. Line 40 has several mechanisms of toxicity reviewed in [5] what does it mean, Please explore it in a better way
  3. Line 54-55 The sentence should be clear; it seems confusing
  4. Line 128-130 The basic should be discussed here and then discussed in the condition of neuro-related diseases
  5. Line 200 Better to include the figures rather than discuss them alone
  6. Line 240, please check the expression of beta, it should be consistent in the whole MS
  7. Comments for Figure 1 The figure requires more improvement; it should be self-explanatory
  8. Line 313 Here, the pathology word was very vast, and the authors could modify with any disorders
  9. Check the representation of Apoe line 328 and 334 and also try to check the whole manuscript; it should be consistent 

Round 2

Reviewer 1 Report

Comments and Suggestions for Authors

I would like to extend my heartfelt thanks to the authors for their thorough and thoughtful revisions. The manuscript has significantly improved since its first submission, demonstrating the authors' sincere dedication to addressing the previously mentioned issues.

Comments on the Quality of English Language

Minor editing of English language required